# Job burnout among Israeli healthcare workers during the first months of COVID-19 pandemic: The role of emotion regulation strategies and psychological distress

**Marlyn Khouri[1] *, Dana Lassri [2,3], Noga Cohen[1,4]**

1 Faculty of Education, University of Haifa, Haifa, Israel, 2 The Paul Baerwald School of Social Work and Social Welfare, The Hebrew University of Jerusalem, Jerusalem, Israel, 3 Research Department of Clinical, Educational and Health Psychology, UCL (University College London), London, United Kingdom, 4 Edmond J. Safra Brain Research Center, University of Haifa, Haifa, Israel

* mkhouri89@gmail.com

**Data Availability Statement:** The dataset generated and analyzed for this study is available on OSF, https://osf.io/h3gdv/.

## Abstract

The current worldwide COVID-19 pandemic has elicited widespread concerns and stress. Arguably, healthcare workers are especially vulnerable to experience burnout during these times due to the nature of their work. Indeed, high prevalence of burnout was found among healthcare workers during the outbreak. However, the individual differences predicting burnout among healthcare workers during the pandemic have been understudied. The aim of the current study was, therefore, to identify risk and protective factors contributing to the severity of burnout among healthcare workers, above and beyond levels of current psychological distress. The survey was distributed online during the period April 13–28, 2020, approximately two months after the first COVID-19 case was identified in Israel. Ninety-eight healthcare workers completed an online survey administered cross-sectionally via the Qualtrics platform that included questionnaires assessing habitual emotion regulation strategies (i.e., trait worry, reappraisal, and suppression), psychological distress, COVID-19 related concerns, and burnout. A hierarchical linear regression analysis revealed that only trait worry and psychological distress were significant predictors of job burnout among healthcare workers. These findings highlight the role of maladaptive emotion regulation tendencies, specifically trait worry, in job burnout among healthcare workers. These findings have implications for both the assessment and treatment of healthcare workers. We discuss potential mechanisms and implications for practice.

## Introduction

The COVID-19 pandemic began in the city Wuhan (Hubei, China) in December 2019 and since then has spread across the globe. The rapid outbreak of the pandemic has led to concerns about personal safety and the safety of close others. The pandemic was defined by the American Psychological Association as an epidemiological and psychological crisis [1], with

**Funding:** The authors received no specific funding for this work.

**Competing interests:** The authors have declared that no competing interests exist.

significant mental health implications both in the present and in the foreseeable future. Accordingly, the pandemic was found to affect individuals' well-being and mental health [2], with the potential for long-lasting effects [2–4]. Indeed, mounting evidence across the globe demonstrated high levels of psychological distress, including post-traumatic stress disorder (PTSD) [5], depression and anxiety [6–8], as well as mood swings, irritability, and sleep disorders [2–4].

Healthcare workers may be especially vulnerable to the psychological toll of pandemics, given the nature of their work ([9]; for review and meta-analysis see [10, 11]). Psychological distress among healthcare workers is a key problem for society, as these professionals provide essential services to the general population. This might be especially important during a pandemic outbreak given the growing number of individuals seeking medical, mental, and welfare assistance, resulting in extreme workload among healthcare workers.

Indeed, high prevalence of psychological distress, such as symptoms of depression, anxiety, and insomnia were found among healthcare workers across the globe during the initial stages of the COVID-19 pandemic [6, 10–13], as well as during prior pandemic outbreaks [14–18]. For example, in recent surveys conducted in China one month after the COVID-19 pandemic outbreak in China, healthcare workers treating patients with COVID-19 exhibited a high prevalence of mental health symptoms, including depression, anxiety, insomnia, and distress [6, 19]. In line with these findings, a large national survey of physicians that was administered in Israel approximately one month following the outbreak of the pandemic found high levels of anxiety among respondents [20]. Similar results were obtained among healthcare workers in Italy showing high prevalence of somatization, distress, sleep disorders and psychopathology symptoms [21, 22]. In Australia, five months after the COVID-19 outbreak, healthcare workers showed significant symptoms of moderate to severe levels of depression, anxiety, and post-traumatic stress disorder [23]. Similarly, in a study conducted during the first months of the pandemic in the United Arab Emirates, over a third of healthcare workers reported moderate or severe psychological distress and moderate or severe anxiety, with more than third of the frontline workers reporting higher levels of anxiety [24]. The COVID-19 pandemic continued to take toll on the mental well-being of healthcare workers also during later stages of the pandemic (September to November 2020), as reflected in high levels of general distress and psychopathology symptoms among this population in various countries (e.g., [25–27]).

These difficulties might be the result of specific characteristics of pandemics, which poses extraordinary amounts of pressure on healthcare workers. According to previous studies conducted in the context of SARS or Ebola epidemics, these situations are characterised by increased workload, physical exhaustion, inadequate personal equipment, and the need to make ethically difficult decisions on the rationing of care [9]. These situational demands may entail dramatic effects on the physical and mental well-being of healthcare workers [9]. Furthermore, pandemics may be associated with experiences of isolation and loss of social support, as well as concerns about risk of infection of friends and relatives [7, 12, 13]. There is strong reasoning to assume that these specific pandemic-related situational characteristics put healthcare workers at high risk for developing psychological distress and experience work-related burnout.

Indeed, in addition to the increase in levels of psychological distress, healthcare workers were found to experience high rates of burnout during the COVID-19 pandemic (e.g., [6, 28]). Burnout has been described as a "state of physical, emotional and mental exhaustion that results from long-term involvement in work situations that are emotionally demanding" [29]. Current notions propose a multidimensional approach that sees burnout as consisting of emotional exhaustion, physical fatigue, and cognitive weariness [30, 31]. Burnout is a key occupational health hazard among healthcare workers. It was found to be associated with depression,

irritability, helplessness, and anxiety among healthcare workers [29]. In addition, burnout has adverse outcomes for the whole organization ([for review see [32]). Thereby, individuals exhibiting burnout demonstrate a reduction in professional performance, greater probability of medical errors, poor patient satisfaction, higher rates of absenteeism, lower commitment to the job and the employer, lower job satisfaction, higher occurrences of medical leave, and greater personal suffering (for review see [32]).

Given the unique nature of the COVID-19 pandemic being both a medical and a psychological crisis [1], healthcare workers were required to work extensively longer hours than usual, both in the health, welfare, and social services systems. It is therefore not surprising that significant levels of burnout were found among healthcare workers worldwide [13, 25, 28, 33, 34], with more than 50% of healthcare workers reporting mild to high levels of burnout during the COVID-19 pandemic [35, 36]. For example, Navarro-Prados et al. [35] found that the rise in working hours in addition to deterioration in mental and physical health contributed to elevation in burnout levels during the pandemic.

Nonetheless, within a similar context, some individuals are more susceptible to burnout than others [37]. Most of the research conducted in the field of professional burnout has been focused on environmental factors [38], as well as on the interactions between environmental and demographic factors [39–42]. However, the research addressing individual differences, such as emotion regulation tendencies that contribute to burnout is relatively sparse [43]. Furthermore, while there is mounting evidence for the rise in burnout levels among healthcare workers during pandemics, and COVID-19 in specific (e.g., [6, 28, 44, 45]), little is known about the risk and protective mechanisms underlying burnout among this population under this specific situational context. In the current work we therefore focused on the role of adaptive and maladaptive emotion regulation strategies in predicting job burnout among this population.

Emotion regulation is defined as the processes by which individuals modify their emotional experiences, expressions, and physiology [46]. Current evidence points at the importance of emotion regulation in predicting one's ability to produce appropriate responses to the ever-changing demands posed by the environment [46]. Such responses may ultimately predict the absence or the presence of psychopathology [46, 47]. In consistence with this idea, difficulties in emotion regulation among healthcare workers are associated with high levels of emotional exhaustion, depersonalization, and lack of personal accomplishment [48], as well as difficulties to verbalize and manage their feelings [49]. Indeed, there is a correlative relationship between emotion regulation tendencies and burnout in healthcare workers [43, 48, 50–53]. However, a recent review depicted that the measures used to assess emotion regulation were highly heterogeneous, making it difficult to draw conclusions about the specific emotion regulation strategies significantly associated with burnout [43]. In the current study, we therefore focused on specific components of emotion regulation, by assessing three main emotion regulation strategies: reappraisal, suppression, and tendency to worry.

Reappraisal is an antecedent focused emotion regulation strategy that involves cognitively construing a potentially emotion-eliciting situation in a way that changes its emotional impact [54]. Reappraisal is effective in reducing negative emotions [55] and has been positively related to psychological wellbeing and health, and negatively related to depressive symptoms [56–58]. A few studies have shown a potential link between reappraisal and burnout among healthcare workers and other professionals such as teachers [59–63], suggesting that a higher tendency to use reappraisal may serve as a protective factor against burnout. Nevertheless, no study to date has examined the association between reappraisal use and burnout during the unique context of intensive stress, as exemplified in the COVID-19 pandemic. This examination is especially important as intensive stress was previously shown to impair individuals' capacity to use reappraisal under intense emotional situations [64], as well as individuals' ability to employ reappraisal [65].

Suppression is a response-focused emotion regulation strategy that involves an active effort to reduce or inhibit the expression of affect after it is aroused [54]. Considerable research has identified suppression as a risk factor for depression [66], anxiety [67, 68], discomfort [69], and heightened sympathetic arousal [70–72]. Previous studies demonstrated the role of suppression in predicting higher levels of perceived stress among healthcare workers [73] and higher levels of burnout among teachers [61]. Nonetheless, while suppression is usually associated with higher levels of psychological distress, there is some evidence that during COVID-19, the ability to suppress emotions is presumably efficient to wellbeing [74]. For example, elevated suppression was found to be associated with lower levels depression, anxiety, and stress [74], as well as with reduced burnout risk among healthcare workers [74]. Therefore, given these contradicting results, further research should assess the role of suppression in predicting job burnout among healthcare workers during the COVID-19 pandemic.

The tendency to worry can be defined as a chain of relatively uncontrollable thoughts that involve attempts to find solutions for an issue whose outcome is uncertain, but may entail negative outcomes [75]. Trait worry is characterized by a reduced tendency to let go of negative feelings [76], and can therefore lead to anxiety, stress, and, depression [77–79]. Chaukos et al. [80] showed higher levels of worry among healthcare residents that have also reported high levels of burnout. This is in line with studies showing a similar relation between worry and burnout among highly demanding professions, such as athletes [81] and university students [82]. Consistently, prior studies have also suggested a strong relationship between trait anxiety and burnout [83–85]. Thus, individuals with high trait anxiety were found to perceive situations in ways that elevate their anxious state, leading eventually to burnout [86].

## The current study

Tying these threads together, the current study aimed to examine whether individual differences in habitual use of emotion regulation strategies play a role in predicting job burnout among healthcare workers in the unique context of the COVID-19 pandemic. Identifying the psychological predictors underlying the vulnerability or resilience to experience burnout may potentially serve as a base for psychosocial interventions for healthcare personnel. As job burnout is closely linked to situational psychological distress [29], especially among populations that experience work-related stress [87], we controlled for episodic COVID-19 related concerns and current psychological distress in our analyses. This allowed us to examine the unique contribution of emotion regulation tendencies to burnout. The following hypotheses were proposed: 1) higher use of reappraisal would be associated with lower levels of burnout; 2) higher levels of suppression would be associated with higher levels of burnout; and 3) higher levels of trait worry would also be associated with higher levels of burnout. We expected to find these results above and beyond levels of current psychological distress.

## Materials and methods

### Participants

Nighty-eight healthcare workers participated in the study (*Mean age* = 36 years, *SD* = 8, 87% female). None of the participants was diagnosed with COVID-19, but nine of the 98 participants reported being in quarantine due to exposure to a person who is carrying the COVID-19, while answering the survey. The study was conducted following APA ethical standards and with approval of the ethics committee of the Faculty of Education at [masked for review]. See Table 1 for demographic characteristics of the sample.

**Table 1. Demographic characteristics of the participants (N = 98).**

| | | n | % |
|---|---|---|---|
| Gender | Male | 13 | 13% |
| | Female | 85 | 87% |
| Education | Undergraduate degree | 45 | 46% |
| | Graduate degree | 53 | 54% |
| Job | Medical staff (doctors, nurses) | 33 | 34% |
| | Paramedical staff (physiotherapists, pharmacists, speech therapists) | 13 | 13% |
| | Clinicians (Psychologists and social workers) | 30 | 31% |
| | Others | 22 | 22% |
| Job experience | Less than one year | 12 | 12% |
| | 1–5 years | 29 | 30% |
| | 5–10 years | 16 | 16% |
| | 10–15 years | 19 | 20% |
| | 15–20 years | 6 | 6% |
| | Above 20 years | 16 | 16% |
| Job percentage | Full time | 64 | 65% |
| | Part time | 18 | 19% |
| | Half time | 8 | 8% |
| | Other | 8 | 8% |
| Salary | Below average | 22 | 22.70% |
| | Average | 34 | 35.10% |
| | Above average | 36 | 37.10% |
| | Far above average | 5 | 5.20% |

## Procedure

Participants completed an online survey administered via the Qualtrics platform (Qualtrics, Provo, UT). The survey was distributed in social media platforms including Facebook groups of healthcare workers, and by emails using a convenience and a snowball sampling (chain-referral sampling). The survey was distributed during the period April 13–28, 2020, approximately two months after the first case of the COVID-19 pandemic was identified in Israel. During this period, the number of people diagnosed with COVID-19 in Israel ranged between 11,586 and 15,728, and the number of deaths ranged between 116 and 210. Legal restrictions on the public included closing schools and entertainment places such as restaurants and shopping centers. The public was asked to maintain social distance by remaining within a 100-meter radius from home, except for essential workers (87% of the individuals in our sample continued going to work). Prior to completing the questionnaire, participants received a detailed explanation regarding the study and signed an informed consent form. The questionnaire took approximately 15 to 20 minutes to complete. The survey included additional questionnaires that are not prominent to the current examination. All data have been made publicly available via the Open Science Framework and can be accessed at https://osf.io/zejwb/.

## Measures

**Demographic questionnaire.** This questionnaire included demographic questions such as age, gender, marital status, religion, job, salary and work experience. Furthermore, this questionnaire included questions related to the COVID-19 pandemic (e.g., "have you been diagnosed with COVID-19?", "Have you recently met someone diagnosed with COVID-19?", "Are you in quarantine?").

**COVID-19 concerns questionnaire (For a similar questionnaire, see [88]).** This questionnaire included questions related to concerns arising from the COVID-19 pandemic: health concerns (own and relatives) and economic concerns (own, relatives and national), as well as concerns about relationships, personal appearance and the ability of the country and the world to cope with the COVID-19 pandemic (e.g., "to what extent do you worry about getting infected with COVID-19?"). Participants responded on a 5-point scale ranging from 1 (not at all) to 5 (very much) (*Cronbach $\alpha$* = .87).

**The Depression Anxiety Stress Scale (DASS; [89]).** The DASS consists of a general factor of psychological distress and orthogonal specific factors of depression, anxiety, and stress [90]. The items on the depression scale include, for example, questions about dysphoria (e.g., "I felt I had lost interest in just about everything") and low self-esteem ("I felt I wasn't worth much as a person"). The items on the anxiety scale assess somatic and subjective responses to anxiety and fear (e.g., "I had difficulty breathing" and "I felt scared without any good reason"). The items on the stress scale measure negative affectivity responses such as nervous tension and irritability, which are characteristic of both depression and anxiety (e.g., "I found it difficult to relax" and "I felt that I was rather touchy"). The respondents rate how often they experienced each item in the past week on a four-point scale ranging from 0 (did not apply to me at all) to 3 (applied to me very much or most of the time). Given a strong correlation between the three subscales (ranging from r = .64 to r = .75), we used the general factor for psychological distress (total DASS score). Cronbach $\alpha$ in our sample for the total DASS score is .93.

**The Penn State Worry Questionnaire (PSWQ; [91]).** The PSWQ is a 16-item measure of trait worry. The items on the scale assess the occurrence, intrusiveness and pervasiveness of worrisome thoughts (e.g., "When I am under pressure I worry a lot"; "My worries overwhelm me"). Participants are instructed to indicate the degree to which each item is typical of them on a five-point scale ranging from 1 = "not at all typical of me" to 5 = "very typical of me". High internal reliability was found between items (*Cronbach $\alpha$* = .84).

**Emotion Regulation Questionnaire (ERQ; [54]).** The ERQ consists of 10 statements that assess two emotion regulation strategies: reappraisal and suppression. Reappraisal is the ability to change the way one thinks about a situation, in order to change how one feels (e.g., "I control my emotions by changing the way I think about the situation I am in"). Suppression is the ability to mask one's feelings and emotional expression (Gross, 1998; e.g., "I control my emotions by not expressing them"). Participants are asked to rate whether they agree or disagree with each statement on a scale of 1 to 7 (1 = strongly disagree, 7 = strongly agree). *Cronbach $\alpha$* = .79 for reappraisal; *Cronbach $\alpha$* = .77 for suppression.

**The Shirom-Melamed Burnout Measure (SMBM; [31]).** The SMBM contains 16 items divided into three subscales: physical fatigue (six items: e.g., "I feel physically drained" or "I feel fed-up"), cognitive weariness (six items: e.g., "I feel I am not thinking clearly" or "I have difficulty concentrating") and emotional exhaustion (four items: e.g., "I feel I am unable to be sensitive to the needs of coworkers and customers" or "I feel I am not capable of being sympathetic to coworkers and customers"). Respondents answer on a seven-point scale ranging from 1 = "almost never" to 5 = "almost always". No significant differences were found between the three factors (physical, emotional, cognitive) and therefore the general score was used. High internal reliability was found between the items (*Cronbach $\alpha$* = .94).

## Analytical strategy

Missing data were replaced with maximum likelihood (ML) estimations based on all variables in the model via Statistical Package for Social Science (SPSS) 27 given that Little's Missing Completely at Random (MCAR) test [92] showed that data were missing completely at

random, $\chi2$ (33) = 21.70, $p$ = .93. Hierarchical linear regression analysis was performed to determine the role of emotion regulation in burnout while controlling for psychological distress, COVID-19 concerns, and demographic variables (age and gender). In step one, age and gender were entered to the model. In step two, psychological distress (total DASS score) and COVID-19 concerns were added. In step three, the emotion regulation strategies (reappraisal, suppression and worry) were added to the model.

## Results

Hierarchical linear regression analysis was performed to identify the significant predictors of burnout among healthcare workers (see Table 2). Burnout scores served as dependent variable. The first step, which included age and gender, did not account for significant variance in burnout, $R^2$ = 0.02, $F(2, 95)$ = .72, $p$ = .49. The second step, in which psychological distress and COVID-19 concerns were added, was significant, $F(4, 93)$ = 15.58, $p < .001$, and accounted for 40% of the variance in burnout. This step added significantly to the model, accounting for an additional 39% of the variance in burnout score, $F$change (2, 93) = 30.00, $p < .001$. In this step, only psychological distress (total DASS score) significantly predicted burnout ($\beta$ = .60, $t$ = 6.21, $p < .001$). The third step, which included also emotion regulation strategies, was also significant, $F(7, 90)$ = 10.97, $p < .001$, and accounted for 46% of the variance in burnout. This step added significantly to the model, accounting for an additional 6% of the variance in burnout score, $F$change (3, 90) = 3.30, p < .05. In this step, only worry ($\beta$ = .24, $t$ = 2.57, $p < .05$) and psychological distress ($\beta$ = .49, $t$ = 4.71, $p < .001$) significantly predicted burnout.

## Discussion

The recent global spread of the coronavirus (COVID-19) has generated a great deal of chaos and stress, particularly among healthcare workers (e.g., [10, 11]). Recent studies identified moderate to severe levels of psychological distress and burnout among healthcare workers in Israel and worldwide [20, 23–25]. The current investigation was conducted in Israel during

**Table 2. Predictors of burnout: Hierarchical regression.**

|  | Predictors | $R^2$ | B | S.E | β | t | p | Confidence Intervals | |
|---|---|---|---|---|---|---|---|---|---|
|  |  |  |  |  |  |  |  | Lower Bound | Upper Bound |
| Step 1 |  | .02 |  |  |  |  |  | 21.49 | 67.29 |
|  | Age |  | -.12 | .19 | -.06 | -.61 | .54 | -.51 | .27 |
|  | Gender |  | 5.30 | 5 | .11 | 1.06 | .29 | -4.63 | 15.23 |
| Step 2 |  | 0.4 |  |  |  |  |  | 15.07 | 60.15 |
|  | Age |  | -.15 | .15 | -.78 | 3.31 | .34 | -.46 | .16 |
|  | Gender |  | 1.47 | 4 | .30 | -.96 | .72 | -6.49 | 9.43 |
|  | COVID-19 Concerns |  | .95 | 1.94 | .05 | .49 | .63 | -2.91 | 4.80 |
|  | Psychological Distress |  | .84 | .14 | .60 | 6.21 | .00 | .57 | 1.11 |
| Step 3 |  | .46 |  |  |  |  |  | -10.38 | 44.26 |
|  | Age |  | -.10 | .15 | -.05 | -.69 | .49 | -.41 | .19 |
|  | Gender |  | .49 | 4.05 | .10 | .12 | .91 | -7.56 | 8.54 |
|  | COVID-19 Concerns |  | .38 | 1.98 | .02 | .19 | .85 | -3.55 | 4.30 |
|  | Psychological Distress |  | .69 | .15 | .49 | 4.71 | .00 | .39 | .98 |
|  | Reappraisal |  | -.07 | .22 | -.02 | -.29 | .77 | -.51 | .38 |
|  | Suppression |  | .46 | .28 | .14 | 1.71 | .09 | -.07 | .99 |
|  | Worry |  | .43 | .17 | .24 | 2.57 | .01 | .09 | .75 |

April 2020, when the number of confirmed cases of COVID-19 rose rapidly, leading to a national lockdown. The main objective of the current study was to examine whether individual differences in habitual use of emotion regulation strategies play a role in predicting job burnout among healthcare workers. We hypothesized that emotion regulation strategies would predict burnout above and beyond levels of psychological distress, COVID-19 concerns, and demographic variables. The results partially support our hypotheses. Specifically, the findings suggest that trait worry and psychological distress were significant predictors of job burnout, whereas habitual reappraisal and suppression did not predict burnout.

Findings from the current study are in agreement with previous results showing an association between levels of worrying and job burnout among healthcare residents [80]. A link between worry and burnout was also found among highly demanding professions, such as athletes [81] and university students [82]. Worry involves concerns about upcoming events and distress that is yet to come. Therefore, given the unique characteristics of COVID-19, including the ambiguity surrounding its nature, the way it spreads, and its treatment, at least in the early stages of the outbreak, it is reasonable to assume that individuals with a high tendency to worry spent many of their time and energy, as well as their cognitive resources on concerns about the future. This might be especially true given their role as healthcare professionals, expected by their patients to provide answers, whilst dealing with their clients' concerns that might actually mirror their own worries. Thereby, presumably further enhancing their worries. Hence, healthcare workers who tend to worry are presumably more vulnerable for accumulating stress, which eventually leads to elevated levels of burnout [93].

While previous studies have exemplified the presence of episodic worries among healthcare workers during COVID-19 pandemic (e.g., [94, 95]), to the best of our knowledge, this study is the first to examine the contribution of *trait worry* to job burnout among healthcare workers. Importantly, in the current study trait worry was a significant predictor of burnout even when controlling for psychological distress and COVID-19 related concerns. This finding suggests that worry is a prominent risk factor for burnout, independently of current symptoms of distress. Therefore, the current investigation adds to the literature on the relationship between burnout and individual differences in emotion regulation (e.g., [50, 53, 96]) by showing that the tendency to worry is associated with job burnout. Therefore, the current findings add to the accumulating evidence regarding the psychological variables that contribute to job burnout [97–100].

Furthermore, our findings are in line with previous studies that demonstrated positive correlations between psychological distress and burnout [101–103]. Recent studies have showed that healthcare workers experience both high levels of psychological distress, as well as fatigue and burnout during the COVID-19 pandemic [13, 22]. An additional study also demonstrated a link between psychological distress and burnout during the pandemic [23]. Levels of psychological distress often have a negative impact on the health, performance, and productivity of workers, what can influence the quality of care provided by them, and ultimately, patients' health [104]. Importantly, although psychological distress was found to be a significant predictor of burnout in our model, levels of worry, a trait-like aspect [91, 105], predicted burnout above and beyond levels of psychological distress.

Surprisingly, the current study did not exemplify a link between the habitual use of reappraisal and burnout levels. This is in contrast to previous studies consistently demonstrating a protective impact of reappraisal on burnout among healthcare workers and other professionals in general [59–63], and in COVID-19 in specific [106]. These findings, however, corroborates the idea that under intense emotional situations, individuals do not tend to rely on reappraisal as a main mechanism to regulate their emotions, but tend to use other emotion regulation strategies such as distraction [64]. Specifically, it may be that the extreme psychological distress

caused by the COVID-19 made reappraisal less effective or accessible [65], thus lowering the potential contribution of reappraisal on lowering job burnout, as showed in previous studies. This suggestion, however, is speculative and should be tested in future studies.

Furthermore, unlike prior findings showing associations between suppression and burnout among teachers and healthcare workers [61, 74], the results of the current study did not support a link between habitual use of suppression and burnout levels. In contrary to previous studies that distinguished between adaptive and maladaptive emotion regulation strategies, to date, researchers emphasize that mental health depends on one's ability to modulate emotional response under situational demands [107–113]. Therefore, adaptation depends on one's ability to flexibly enhance or suppress emotional reactions in accordance with the contextual demands [111]. Indeed, in a recent study conducted in Italy by Lenzo et al. [114], it was found that the ability to flexibility enhance or suppress emotional response decreased burnout risk in the context of palliative home care [114].

## Limitations

Findings of this study should be evaluated in the light of several methodological limitations. The sample size was relatively small, predominantly female, which likely limited the generalizability of results. In addition, the current study was cross-sectional in nature, which does not allow for examining directionality between variables. Prospective research with larger samples is needed to untangle directionality of current findings. Although all study measures were assessed via the use of commonly and validated questionnaires, it is possible that they represent self-perception and conscious subjective processes.

## Implications

Findings of this study may contribute to the understanding of the psychological dynamics accruing in the context of an epidemiological and psychological crisis and have important implications. Taken together, the results of the present study provide further knowledge regarding the risk factors that might enhance healthcare workers' vulnerability to professional burnout at times of crisis. Thus, emphasize the need for novel interventions for preventing psychological distress and promoting well-being among populations whose ability to function is crucial at times of crisis. Our results suggest that in order to reduce the risk of burnout and improve the well-being of healthcare workers, intervention planning should integrate programs that address Person-centered interventions aimed at reducing the tendency to worry [115].

## Author Contributions

**Conceptualization:** Marlyn Khouri, Noga Cohen.

**Formal analysis:** Marlyn Khouri, Dana Lassri.

**Investigation:** Marlyn Khouri.

**Methodology:** Marlyn Khouri, Noga Cohen.

**Supervision:** Noga Cohen.

**Writing – review & editing:** Marlyn Khouri, Dana Lassri, Noga Cohen.

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
