## [Decision Letter · Decision Letter 0]

13 Dec 2021

PONE-D-21-27667Job Burnout among Israeli Healthcare Workers during COVID-19 Pandemic: The Role of Emotion Regulation Strategies and Psychological DistressPLOS ONE

Dear Dr. Cohen,

Thank you for submitting your manuscript to PLOS ONE and apologies for the delay in our response. Due to the pandemic it was difficult to find reviewers. After careful consideration, we feel that it has merit but does not fully meet PLOS ONE’s publication criteria as it currently stands. Therefore, we invite you to submit a revised version of the manuscript that addresses the points raised during the review process. Both reviewers have provided comments that need to be addressed.

We look forward to receiving your revised manuscript.

Kind regards,

Therese van Amelsvoort

Academic Editor

PLOS ONE

“All authors have no conflict of interest to disclose”

Reviewers' comments:

Reviewer's Responses to Questions

**Comments to the Author**

1. Is the manuscript technically sound, and do the data support the conclusions?

Reviewer #1: Yes

Reviewer #2: Yes

2. Has the statistical analysis been performed appropriately and rigorously? 

Reviewer #1: No

Reviewer #2: Yes

3. Have the authors made all data underlying the findings in their manuscript fully available?

Reviewer #1: Yes

Reviewer #2: Yes

4. Is the manuscript presented in an intelligible fashion and written in standard English?

Reviewer #1: Yes

Reviewer #2: Yes

5. Review Comments to the Author

Reviewer #1: This manuscript “Job Burnout among Israeli Healthcare Workers during the COVID-19 Pandemic: The Role of Emotion Regulation Strategies and Psychological Distress” describes the results of a cross-sectional investigation of work burnout and emotion regulation strategies, as well as worries, depression, anxiety and stress, among Israeli healthcare workers (HCW) during the initial stage of the COVID-19 pandemic in Israel. The researchers found associations among burnout with the rest of the study variables, except for reappraisal. I believe that this is an important study which can add to the relatively less-investigated body of knowledge regarding emotion regulation strategies of HCW during COVID-19. Yet, I do have some concerns which I believe should be addressed in order to improve this paper:

In my opinion, the weakness of this manuscript is its analytic approach. The choice to conduct a stepwise regression model is unclear to me from several reasons: first, this is an exploratory approach in its nature, and as such it suffers from all the pitfalls of fishing expedition. Moreover, the choice of this method is surprising since the researchers have some very clear hypotheses, which are not being addressed properly this way. As a result- the researchers’ interpretation of their results is incorrect. Stepwise does not locate the most prominent predictors of a dependent variable, but rather creates the best combination of factors, but on the way some extremely important factors are being omitted (for example- anxiety, stress and the ERQ subscales). If the authors wish to avoid multicollinearity there are other ways- from using the standardized residuals of regressed independent variables, to combining highly correlated variables or even conducting a series of simple regressions with alpha corrections.

More importantly, there are several factors that simply cannot be excluded from the model: the first are age and sex (which is crucial since the sample is biased in favor of women). The second is the emotion regulation strategies, which are the focus of the paper! I suggest that the authors will conduct a standard regression model with the fixed factors that were decided a-priori, and will address the issue of multicollinearity in a different way. Of course, if you decide to do so the discussion should be adjusted to your renewed results. Also, there is no need to conduct both regression and correlations, so I believe that the authors should focus on one model with their variables of interest.

On a similar note, I think that some of the information presented in Table 3 is redundant. For example- there is no need to present both tolerance and VIF (which is calculated as 1/tolerance). A simple report of the VIF is enough (and to my taste even this is unnecessary unless there is an issue of multicollinearity in the model itself). The presentation of the confidence interval (upper and lower limit) in unstandardized scores is not informative, and it is much preferred if it would be presented in standardized scores (as the second beta column). The constant is not interesting and can be hidden from the table. Also, I don’t see the point of presenting the excluded factors in such a detailed manner. In Table 2- notice that you are not consistent with the number of decimal points (.296 vs. .64 etc.).

Some additional notes:

Introduction:

1. The review of the situation of HCW during COVID-19 is a bit lack. Many papers have been published since the outbreak and there are many reviews, meta-analyses and high-quality surveys on the experiences of HCW. I think that it’s a shame to address in such details to three specific surveys (page 4 lines 67-75) that were published very close to the first emergence of the outbreak and are not representative to the Israeli population nor do they capture a global snapshot of the situation, especially if one is from MedRxiv and haven’t been peer-reviewed.

2. Page 4 line 89- not “during pandemics” but “during the COVID-19 pandemic”

3. Page 5 line 114- you claim that “the research addressing individual differences that may potentially contribute to burnout is relatively sparse”, please provide a reference to support this claim. Maybe a review or meta-analysis of some kind. To the best of my knowledge there have been numerous investigations of risk factors for burnout among HCW, but your uniqueness is the focus on emotion regulation.

4. Page 8 line 188- not “personal” but “personnel”.

Methods:

1. Under procedure- your survey was not conducted “approximately one month after the outbreak of the COVID-19 pandemic in Israel”, as claimed, but rather almost two months after the first case was discovered in late February, and almost one month into the national lockdown.

2. The Cronbach’s alphas for the ERQ are only moderate, is it similar to the questionnaire’s psychometrics in other studies? If not- can you offer an explanation for this?

Discussion:

1. Page 13 line 297- the lockdown in Israel was by no way partial, but rather a full lockdown in which only a small group of essential workers was excluded.

2. “in the stepwise regression analysis only the tendency to worry and levels of depression remained significant predictors of burnout”- As mentioned earlier, this interpretation is incorrect. The variables should be examined in a fixed model which includes all of the predictors in order to determine their relationship with burnout and one another.

“Habitual reappraisal, however, was not linked to burnout”- this conclusion is drawn from the correlations, but again- in a standard regression you can include it and reach this conclusion without the need to run both regression and correlations.

3. Page 14 line 316- write “patients” instead of “clients”

4. Page 14 second paragraph (“Importantly, in the current study…”)- again I believe that you interpret exploratory results in an incorrect way, and claim that worry predicted burnout “independently of anxiety and stress”, even though they were not included in the model! Please revise this passage in accordance with your actual results.

5. Page 14 lines 305-319- I find it peculiar that you do not address any findings from COVID-19. There are tons of recent evidence that link to your findings, which I believe should be discussed here.

6. There are a few grammatical errors throughout the paper (“although depression was found a significant predictor”, instead of “was found to be a…”). Please revise the text and address them.

7. Page 14 line 337- you claim that worries are a trait-like aspect while depression is more episodic- please provide a reference to support this claim. Otherwise, one might insist that worries are also an episodic state, a claim which sounds reasonable to me.

8. Limitations- power is not one of your limitations, despite your small sample size. It is visible in table 2 that where a respectable effect size was detected, the result was statistically significant. Therefore, I do not believe that insignificant results in your study ensue from type-II errors.

To conclude, I honestly believe that this study has a lot of potential and is of great value, but the interesting data presented here should be analyzed and interpreted a bit more rigorously in order to draw reliable and accurate conclusions. I am looking forward to reading the revised version of the manuscript.

Reviewer #2: This is my review on the manuscript entitled “Job Burnout among Israeli Healthcare Workers during the COVID-19 Pandemic: The Role of Emotion Regulation Strategies and Psychological Distress”.

Burn out is a serious problem with devastating effects not only on healthcare workers but also to the health system as well. Thus, this study is important and provides significant information regarding the main risk and protective factors that contributes to burn out and psychological deterioration.

The current investigation took place 3 months after the start of the pandemic and one month after the declaration of the Covid-19 situation as a pandemic by the WHO (March 11th, 2020). Since it is almost 2 years after the appearance of Covid-29 in Wuhan, I recommend that the title should be modified accordingly. For instance, “Job Burnout among Israeli Healthcare Workers during the first months of COVID-19 Pandemic: The Role of Emotion Regulation Strategies and Psychological Distress”.

The introduction is well written. Materials and methods are sufficient. You should mention the population target as you distributed the survey through Facebook and emails (not through a collective source like hospital emails etc.). Was it accessible by everyone from Facebook? Was it through a Healthcare workers Facebook group, or personal message? Statistics are fine. Table 1 needs to be presented with the number of each demographic characteristic alongside with percent.

Discussion is interesting. You should include a small paragraph with a country-based comparison to Israeli Healthcare workers by including these studies:

1. Dobson H, et al. Burnout and psychological distress amongst Australian healthcare workers during the COVID-19 pandemic. Australas Psychiatry. 2021 Feb;29(1):26-30. doi: 10.1177/1039856220965045. Epub 2020 Oct 12. Erratum in: Australas Psychiatry. 2021 May 11;:10398562211011741. PMID: 33043677; PMCID: PMC7554409.

2. Cheristanidis S, et al. Psychological Distress in Primary Healthcare Workers during the COVID-19 Pandemic in Greece. Acta Med Acad. 2021 Aug;50(2):252-263. doi: 10.5644/ama2006-124.341. PMID: 34847678.

3. Saddik B, et al. Psychological Distress and Anxiety Levels Among Health Care Workers at the Height of the COVID-19 Pandemic in the United Arab Emirates. Int J Public Health. 2021 Nov 11;66:1604369. doi: 10.3389/ijph.2021.1604369. PMID: 34840553; PMCID: PMC8615074.

4. Firew T, et al. Protecting the front line: a cross-sectional survey analysis of the occupational factors contributing to healthcare workers' infection and psychological distress during the COVID-19 pandemic in the USA. BMJ Open. 2020 Oct 21;10(10):e042752. doi: 10.1136/bmjopen-2020-042752. PMID: 33087382; PMCID: PMC7580061.

Language is good.

6. PLOS authors have the option to publish the peer review history of their article (what does this mean?). If published, this will include your full peer review and any attached files.

Reviewer #1: **Yes: **Nimrod Hertz-Palmor

Reviewer #2: No

---

## [Author Response · Author response to Decision Letter 0]

2 Feb 2022

Thank you for your e-mail offering us the opportunity to revise and resubmit our paper entitled “Job Burnout Among Israeli Healthcare Workers During First Months of COVID-19 Pandemic: The Role of Emotion Regulation Strategies and Psychological Distress” to PLOS ONE. We found the reviewers’ comments highly informative and helpful, and below we have specified how we have addressed each of them.

We believe that the manuscript is much improved following the reviewers’ comments and the changes we made. In the revised manuscript, the changes that have been made are marked.

Comments of Reviewer #1

This manuscript “Job Burnout among Israeli Healthcare Workers during the COVID-19 Pandemic: The Role of Emotion Regulation Strategies and Psychological Distress” describes the results of a cross-sectional investigation of work burnout and emotion regulation strategies, as well as worries, depression, anxiety and stress, among Israeli healthcare workers (HCW) during the initial stage of the COVID-19 pandemic in Israel. The researchers found associations among burnout with the rest of the study variables, except for reappraisal. I believe that this is an important study which can add to the relatively less-investigated body of knowledge regarding emotion regulation strategies of HCW during COVID-19. Yet, I do have some concerns which I believe should be addressed in order to improve this paper:

Response: We thank the reviewer for pointing out the strengths of our work and for his insightful comments.

In my opinion, the weakness of this manuscript is its analytic approach. The choice to conduct a stepwise regression model is unclear to me from several reasons: first, this is an exploratory approach in its nature, and as such it suffers from all the pitfalls of fishing expedition. Moreover, the choice of this method is surprising since the researchers have some very clear hypotheses, which are not being addressed properly this way. As a result- the researchers’ interpretation of their results is incorrect. Stepwise does not locate the most prominent predictors of a dependent variable, but rather creates the best combination of factors, but on the way some extremely important factors are being omitted (for example- anxiety, stress and the ERQ subscales). If the authors wish to avoid multicollinearity there are other ways- from using the standardized residuals of regressed independent variables, to combining highly correlated variables or even conducting a series of simple regressions with alpha corrections.

More importantly, There are several factors that simply cannot be excluded from the model: the first are age and sex (which is crucial since the sample is biased in favor of women). The second is the emotion regulation strategies, which are the focus of the paper! I suggest that the authors will conduct a standard regression model with the fixed factors that were decided a-priori, and will address the issue of multicollinearity in a different way. Of course, if you decide to do so the discussion should be adjusted to your renewed results. Also, there is no need to conduct both regression and correlations, so I believe that the authors should focus on one model with their variables of interest.

Response: We thank the reviewer for this important comment. Following the reviewer's advice, we have now calculated a total score for the DASS questionnaire, which represents overall psychological distress, instead of entering the different subscales into the analysis. This idea is consistent with previous works that look at the total DASS score (e.g., Lamuri et al., 2021). In addition, we omitted the correlation table from the manuscript. Furthermore, we changed the analytical plan and now report the results of a hierarchical linear regression instead of the stepwise regression. The new analysis consists of three steps: In the first step, in line with the reviewer's suggestion, we included age and gender in the model. In the second step, we added total DASS scores and COVID-related concerns. In the third step, as the reviewer suggested, we added the emotion regulation measures (suppression, reappraisal, and worry). The pattern of results remained consistent with the previous results, showing that only worry and psychological distress predict job burnout, even when controlling for age and gender.

On a similar note, I think that some of the information presented in Table 3 is redundant. For example- there is no need to present both tolerance and VIF (which is calculated as 1/tolerance). A simple report of the VIF is enough (and to my taste even this is unnecessary unless there is an issue of multicollinearity in the model itself). The presentation of the confidence interval (upper and lower limit) in unstandardized scores is not informative, and it is much preferred if it would be presented in standardized scores (as the second beta column). The constant is not interesting and can be hidden from the table. Also, I don’t see the point of presenting the excluded factors in such a detailed manner. 

Response: We thank the reviewer for pointing this out. After combining the different DASS subscales into a single measure there was no multicollinearity. Therefore, we do not report VIF or tolerance. 

In Table 2- notice that you are not consistent with the number of decimal points (.296 vs. .64 etc.). 

Response: We made sure that all values contain 2 decimal points.

The review of the situation of HCW during COVID-19 is a bit lack. Many papers have been published since the outbreak and there are many reviews, meta-analyses and high-quality surveys on the experiences of HCW. I think that it’s a shame to address in such details to three specific surveys (page 4 lines 67-75) that were published very close to the first emergence of the outbreak and are not representative to the Israeli population nor do they capture a global snapshot of the situation, especially if one is from MedRxiv and haven’t been peer-reviewed. 

Response: We have updated the literature review and added investigations conducted both immediately and later on during the COVID-19 pandemic (page 3-5, lines 62-84).

Page 4 line 89- not “during pandemics” but “during the COVID-19 pandemic”. 

Response: This sentence has been modified.

Page 5 line 114- you claim that “the research addressing individual differences that may potentially contribute to burnout is relatively sparse”, please provide a reference to support this claim. Maybe a review or meta-analysis of some kind. To the best of my knowledge, there have been numerous investigations of risk factors for burnout among HCW, but your uniqueness is the focus on emotion regulation. 

Response: Indeed, there are numerous investigations addressing personal factors predicting burnout among healthcare workers. We have added references regarding environmental and personal factors that contribute to burnout (page 6, lines 122-134). We also added a sentence clarifying that the research addressing the emotion regulation tendencies that may potentially contribute to burnout is relatively sparse.

Page 8 line 188- not “personal” but “personnel”. 

Response: Thank you. we changed the word accordingly.

Under procedure- your survey was not conducted “approximately one month after the outbreak of the COVID-19 pandemic in Israel”, as claimed, but rather almost two months after the first case was discovered in late February, and almost one month into the national lockdown. 

Response: We thank the reviewer for this clarification. We have changed the text accordingly (page 10, lines 223-224).

The Cronbach’s alphas for the ERQ are only moderate, is it similar to the questionnaire’s psychometrics in other studies? If not- can you offer an explanation for this?

Response: The Cronbach’s alphas reliabilities (.79 for reappraisal and .77 for suppression) we received in our sample are similar to those found in prior research (e.g., Gross and John, 2003; range between .75 to .82 for reappraisal and range between .68 to .76 for suppression). It seems that in the previous version, there was a typo in the reliability written for suppression (which is .77 and not .73). We apologize for that. 

Page 13 line 297- the lockdown in Israel was by no way partial, but rather a full lockdown in which only a small group of essential workers was excluded. 

Response: Thank you. We have changed the text accordingly.

 “In the stepwise regression analysis only the tendency to worry and levels of depression remained significant predictors of burnout”- As mentioned earlier, this interpretation is incorrect. The variables should be examined in a fixed model which includes all of the predictors in order to determine their relationship with burnout and one another. “Habitual reappraisal, however, was not linked to burnout”- this conclusion is drawn from the correlations, but again- in a standard regression you can include it and reach this conclusion without the need to run both regression and correlations.

Response: As noted above, we have changed the analytic strategy in line with the reviewer's suggestions.

Page 14 line 316- write “patients” instead of “clients”. 

Response: Ass suggested, we have replaced the word "clients" with the word "patient".

Page 14 second paragraph (“Importantly, in the current study…”)- again I believe that you interpret exploratory results in an incorrect way, and claim that worry predicted burnout “independently of anxiety and stress”, even though they were not included in the model! Please revise this passage in accordance with your actual results.

Response: As mentioned earlier, we conducted a hierarchical regression analysis. This analysis showed that worry predicts burnout above and beyond psychological distress.

Page 14 lines 305-319- I find it peculiar that you do not address any findings from COVID-19. There are tons of recent evidence that link to your findings, which I believe should be discussed here.

Response: Although previous research emphasized the role of worries related to the COVID-19 pandemic, our study is the first, to our knowledge, that looked at worry as a trait characteristic. Nevertheless, we now included papers on COVID-19 related worries and clarified the unique contribution of the current investigation. 

There are a few grammatical errors throughout the paper (“although depression was found a significant predictor”, instead of “was found to be a…”). Please revise the text and address them.

Response: Thank you. The manuscript was sent for an English editing before submission. 

Page 14 line 337- you claim that worries are a trait-like aspect while depression is more episodic- please provide a reference to support this claim. Otherwise, one might insist that worries are also an episodic state, a claim which sounds reasonable to me.

Response: In the current study we chose to use a questionnaire assessing worry as a trait characteristic (i.e., PSWQ; Meyer et al., 1990). In order to assess worries specifically related to COVID-19 (and therefore episodic in nature) we also included a COVID-19 concerns questionnaire. We clarified in the text that worry was assessed as a trait factor. As can be seen from the results, trait worry predicted burnout above and beyond levels of episodic (COVID-19-related) concerns. 

Limitations- power is not one of your limitations, despite your small sample size. It is visible in table 2 that where a respectable effect size was detected, the result was statistically significant. Therefore, I do not believe that insignificant results in your study ensue from type-II errors.

Response: We changed the sentence about power according to the suggestion.

To conclude, I honestly believe that this study has a lot of potential and is of great value, but the interesting data presented here should be analyzed and interpreted a bit more rigorously in order to draw reliable and accurate conclusions. I am looking forward to reading the revised version of the manuscript.

Response: We wish to thank the reviewer for emphasizing the importance of our work and for his insightful comments.

Comments of Reviewer #2: 

This is my review on the manuscript entitled “Job Burnout among Israeli Healthcare Workers during the COVID-19 Pandemic: The Role of Emotion Regulation Strategies and Psychological Distress”. Burnout is a serious problem with devastating effects not only on healthcare workers but also to the health system as well. Thus, this study is important and provides significant information regarding the main risk and protective factors that contributes to burn out and psychological deterioration. The current investigation took place 3 months after the start of the pandemic and one month after the declaration of the Covid-19 situation as a pandemic by the WHO (March 11th, 2020). Since it is almost 2 years after the appearance of Covid-29 in Wuhan, I recommend that the title should be modified accordingly. For instance, “Job Burnout among Israeli Healthcare Workers during the first months of COVID-19 Pandemic: The Role of Emotion Regulation Strategies and Psychological Distress”.

Response: We thank the reviewer for highlighting the importance of the current work and for the helpful comments. We changed the title according to the reviewer's suggestion.

The introduction is well written. Materials and methods are sufficient. 

Response: Many thanks.

You should mention the population target as you distributed the survey through Facebook and emails (not through a collective source like hospital emails etc.). Was it accessible by everyone from Facebook? Was it through a Healthcare workers Facebook group, or personal message? 

Response: The survey was distributed in social media platforms including Facebook groups of Healthcare workers and via snowball sampling (chain-referral sampling). We have now included this information in the text (page 10, lines 220-225).

Statistics are fine. Table 1 needs to be presented with the number of each demographic characteristic alongside with percent.

Response: We have added this information to the table.

Discussion is interesting. You should include a small paragraph with a country-based comparison to Israeli Healthcare workers by including these studies:

1. Dobson H, et al. Burnout and psychological distress amongst Australian healthcare workers during the COVID-19 pandemic. Australas Psychiatry. 2021 Feb;29(1):26-30. doi: 10.1177/1039856220965045. Epub 2020 Oct 12. Erratum in: Australas Psychiatry. 2021 May 11;10398562211011741. PMID: 33043677; PMCID: PMC7554409.

2. Cheristanidis S, et al. Psychological Distress in Primary Healthcare Workers during the COVID-19 Pandemic in Greece. Acta Med Acad. 2021 Aug;50(2):252-263. doi: 10.5644/ama2006-124.341. PMID: 34847678.

3. Saddik B, et al. Psychological Distress and Anxiety Levels Among Health Care Workers at the Height of the COVID-19 Pandemic in the United Arab Emirates. Int J Public Health. 2021 Nov 11;66:1604369. doi: 10.3389/ijph.2021.1604369. PMID: 34840553; PMCID: PMC8615074.

4. Firew T, et al. Protecting the front line: a cross-sectional survey analysis of the occupational factors contributing to healthcare workers' infection and psychological distress during the COVID-19 pandemic in the USA. BMJ Open. 2020 Oct 21;10(10):e042752. doi: 10.1136/bmjopen-2020-042752. PMID: 33087382; PMCID: PMC7580061.

Response: We thank the reviewer for providing these references. We have updated the literature review and added a 

paragraph in the introduction discussing distress of HCW in different countries.

Language is good.

Response: Thank you.

---

## [Decision Letter · Decision Letter 1]

21 Feb 2022

PONE-D-21-27667R1Job Burnout Among Israeli Healthcare Workers During the First Months of COVID-19 Pandemic: The Role of Emotion Regulation Strategies and Psychological DistressPLOS ONE

Dear Dr. Cohen,

Thank you for submitting your manuscript to PLOS ONE. The reviewers feel the manuscript has improved, however there are still some minor points remaining. Therefore, we invite you to submit a revised version of the manuscript that addresses the points raised during the review process.

We look forward to receiving your revised manuscript.

Kind regards,

Therese van Amelsvoort

Academic Editor

PLOS ONE

Journal Requirements:

Reviewers' comments:

Reviewer's Responses to Questions

**Comments to the Author**

1. If the authors have adequately addressed your comments raised in a previous round of review and you feel that this manuscript is now acceptable for publication, you may indicate that here to bypass the “Comments to the Author” section, enter your conflict of interest statement in the “Confidential to Editor” section, and submit your "Accept" recommendation.

Reviewer #1: (No Response)

Reviewer #2: All comments have been addressed

2. Is the manuscript technically sound, and do the data support the conclusions?

Reviewer #1: Yes

Reviewer #2: Yes

3. Has the statistical analysis been performed appropriately and rigorously? 

Reviewer #1: Yes

Reviewer #2: Yes

4. Have the authors made all data underlying the findings in their manuscript fully available?

Reviewer #1: Yes

Reviewer #2: Yes

5. Is the manuscript presented in an intelligible fashion and written in standard English?

Reviewer #1: Yes

Reviewer #2: Yes

6. Review Comments to the Author

Reviewer #1: I thank the authors for taking my comments into consideration and for revising the manuscript so thoroughly. I believe the current version is much improved. A few minor comments about the results section:

1. Please refer to R square as 0.02, and not 2%.

2. When reporting the steps of the hierarchical regression, please include the R square change value, which is crucial to get an idea about your effect size.

I have no other comments beyond that

Reviewer #2: (No Response)

7. PLOS authors have the option to publish the peer review history of their article (what does this mean?). If published, this will include your full peer review and any attached files.

Reviewer #1: **Yes: **Nimrod Hertz-Palmor

Reviewer #2: No

---

## [Author Response · Author response to Decision Letter 1]

26 Feb 2022

I thank the authors for taking my comments into consideration and for revising the manuscript so thoroughly. I believe the current version is much improved. A few minor comments about the results section:

1. Please refer to R square as 0.02, and not 2%.

2. When reporting the steps of the hierarchical regression, please include the R square change value, which is crucial to get an idea about your effect size. 

Response: We thank the reviewer for the time and effort he invested in reviewing our manuscript. We changed the R2 value to 0.02 and also added details about R2 change.

Page 13: “Burnout scores served as dependent variable. The first step, which included age and gender, did not account for significant variance in burnout, R2 = 0.02, F(2, 95) = .72, p = .49. The second step, in which psychological distress and COVID-19 concerns were added, was significant, F(4, 93) = 15.58, p < .001, and accounted for 40% of the variance in burnout. This step added significantly to the model, accounting for an additional 39% of the variance in burnout score, Fchange (2, 93) = 30.00, p < .001. In this step, only psychological distress (total DASS score) significantly predicted burnout (β = .60, t = 6.21, p < .001). The third step, which included also emotion regulation strategies, was also significant, F(7, 90) = 10.97, p < .001, and accounted for 46% of the variance in burnout. This step added significantly to the model, accounting for an additional 6% of the variance in burnout score, Fchange (3, 90) = 3.30, p < .05. In this step, only worry (β = .24, t = 2.57, p < .05) and psychological distress (β = .49, t = 4.71, p < .001) significantly predicted burnout.”

---

## [Decision Letter · Decision Letter 2]

7 Mar 2022

Job Burnout Among Israeli Healthcare Workers During the First Months of COVID-19 Pandemic: The Role of Emotion Regulation Strategies and Psychological Distress

PONE-D-21-27667R2

Dear Dr. Cohen,

We’re pleased to inform you that your manuscript has been judged scientifically suitable for publication and will be formally accepted for publication once it meets all outstanding technical requirements.

Kind regards,

Therese van Amelsvoort

Academic Editor

PLOS ONE

Additional Editor Comments (optional):

Reviewers' comments:

Reviewer's Responses to Questions

**Comments to the Author**

1. If the authors have adequately addressed your comments raised in a previous round of review and you feel that this manuscript is now acceptable for publication, you may indicate that here to bypass the “Comments to the Author” section, enter your conflict of interest statement in the “Confidential to Editor” section, and submit your "Accept" recommendation.

Reviewer #1: All comments have been addressed

2. Is the manuscript technically sound, and do the data support the conclusions?

Reviewer #1: Yes

3. Has the statistical analysis been performed appropriately and rigorously? 

Reviewer #1: Yes

4. Have the authors made all data underlying the findings in their manuscript fully available?

Reviewer #1: Yes

5. Is the manuscript presented in an intelligible fashion and written in standard English?

Reviewer #1: Yes

6. Review Comments to the Author

Reviewer #1: (No Response)

7. PLOS authors have the option to publish the peer review history of their article (what does this mean?). If published, this will include your full peer review and any attached files.

Reviewer #1: **Yes: **Nimrod Hertz-Palmor

---

## [Editor Report · Acceptance letter]

15 Mar 2022

PONE-D-21-27667R2 

Job Burnout Among Israeli Healthcare Workers During the First Months of COVID-19 Pandemic: The Role of Emotion Regulation Strategies and Psychological Distress 

Dear Dr. Cohen:

I'm pleased to inform you that your manuscript has been deemed suitable for publication in PLOS ONE. Congratulations! Your manuscript is now with our production department. 

Kind regards, 

on behalf of

Prof. Therese van Amelsvoort 

Academic Editor

PLOS ONE